# The Efficacy of Faecal Microbiota Transplant and Rectal Bacteriotherapy in Patients with Recurrent *Clostridioides difficile* Infection: A Retrospective Cohort Study

**DOI:** 10.3390/cells11203272

**Published:** 2022-10-18

**Authors:** Camilla Kara Svensson, Frederik Cold, Iben Ribberholt, Mike Zangenberg, Hengameh Chloé Mirsepasi-Lauridsen, Andreas Munk Petersen, Morten Helms

**Affiliations:** 1Department of Infectious Diseases, Copenhagen University Hospital Hvidovre, 2650 Hvidovre, Denmark; 2Gastro Unit, Medical Division, Copenhagen University Hospital Hvidovre, 2650 Hvidovre, Denmark; 3Department of Plant and Environmental Sciences, Copenhagen University, 1871 Copenhagen, Denmark; 4Department of Clinical Microbiology, Copenhagen University Hospital Hvidovre, 2650 Hvidovre, Denmark; 5Department of Clinical Medicine, Faculty of Health and Medical Sciences, Copenhagen University, 2100 Copenhagen, Denmark

**Keywords:** *Clostridioides difficile* infection, CDI, faecal microbiota transplantation, FMT, rectal bacteriotherapy, RBT

## Abstract

The most effective treatment for recurrent *Clostridioides difficile* infection (CDI) is faecal microbiota transplantation (FMT); however, the optimal route of administration is thus far unknown. This retrospective cohort study of 343 patients sought to evaluate the efficacy of treatment with FMT capsules, FMT enema, and rectal bacteriotherapy (RBT) during a five-year period. The primary endpoint was clinical resolution from CDI after eight weeks, and secondary endpoints were time to recurrence and death during the follow-up period. The proportion of patients with clinical resolution was 79.9% in the FMT capsule group, 53.3% in the FMT enema group, and 61.8% in the RBT group, corresponding to an adjusted odds ratio of 3.79 (CI: 1.82 to 8.26) in the FMT capsule group compared with FMT enema, and 2.92 (CI: 1.49 to 6.03) compared with RBT. The hazards ratio for recurrence within the first 12 months of follow-up was 0.24 (CI: 0.06 to 0.89) in the FMT capsule group compared with FMT enema, and 0.26 (CI: 0.08 to 0.91) compared with RBT. There was no difference in mortality. In conclusion, FMT capsules were more effective than both FMT enema and RBT as treatment of recurrent CDI and reduced the risk of further recurrences.

## 1. Introduction

*Clostridioides difficile* is an anaerobic, Gram-positive, toxin-producing bacteria, and a well-established cause of nosocomial diarrhoea [1,2]. Infection with *Clostridioides difficile* is associated with increased morbidity and mortality [3,4], as well as an economic burden to the health care system [5,6]. Despite standard-care treatment, 25–60% of patients experience recurrence of *Clostridioides difficile* infections (CDI) [7].

Current treatment modalities for recurrent CDI (rCDI) are vancomycin, fidaxomicin and/or Faecal Microbiota Transplantation (FMT). FMT preceded by antibiotic treatment being the most effective treatment option [8,9,10] with cure rates between 60% and 90% and reduced risk of new recurrences [11]. However, the production method and mode of delivery of FMT varies considerably between studies, and the optimal route of administration is so far unknown [12,13,14].

Encapsulated FMT is easily administered compared with rectal administration and is a highly effective treatment option with cure rates comparable to other modes of delivery [15].

At Copenhagen University Hospital Hvidovre (CUHH), FMT treatment is managed in the Department of Infectious Diseases, whereas other Danish FMT centres are centred at Departments of Gastroenterology. Therefore, FMT via endoscopy is rarely offered in CUHH, whereas it is more routinely used in the other centres [16].

At CUHH an alternative treatment option, rectal bacteriotherapy (RBT), has been applied. RBT consists of a standardised bacterial mixture of 12 well-defined bacteria strains [17,18].

One randomised clinical trial reported no difference in efficacy between RBT and FMT enema of 52–56% following a single treatment (one day of FMT vs. three consecutive days of RBT) [19]; however, the study population was small.

The safety of RBT and FMT has been reported equal [20]. As the bacterial mixture is produced in a laboratory, RBT is not reliant on donor material, and production of RBT might therefore be logistically easier. The aim of the study was to investigate and compare the effect of the three treatment modalities FMT capsules, FMT enema, and RBT on recurrent or treatment refractory CDI in a retrospective cohort study.

## 2. Materials and Methods

The study was conducted as part of an internal quality assessment and approved by the hospital directory board at Copenhagen University Hospital Hvidovre (CUHH) (wz19001024-2019-62).

### 2.1. Study Population

All patients from the Capital Region of Denmark (population approximately 1.8 million) presenting with recurrent or refractory CDI and for whom FMT or RBT were considered treatment options, were referred to the Department of Infectious Diseases at CUHH. All patients treated with FMT or RBT in the period 1 January 2017 to 31 December 2021 were screened for inclusion in this study. The last follow up was 1 March 2022. Data were obtained from patients’ electronic medical records.

Inclusion criteria were age ≥ 18 years, recurrent CDI, refractory or severe/fulminant CDI regardless of the number previous CDI episodes (as defined in the Danish national guidelines [21]), and treatment with either FMT or RBT.

Only one patient in the five-year period was treated via upper endoscopy and was therefore excluded. No patients were treated with FMT via the nasogastric/jejunal route.

Other exclusion criteria were treatment with FMT or RBT for other conditions, or treatment with FMT or RBT in the previous six months. FMT or RBT administered during participation in clinical trials or treatment with FMT capsules produced at the private hospital Aleris-Hamlet were also excluded, as these results are reported elsewhere [19,22] (Figure 1). See Appendix A for definitions.

### 2.2. Demographics and Clinical Information

The following data were assessed; age, gender, Charlson Comorbidity Index (CCI), [23,24] immunosuppressive medication or disease, treatment with proton-pump inhibitors, previous FMT or RBT, number of previous CDI episodes, recurrent or refractory CDI, severity of current CDI episode, *Clostridioides difficile* subtype, date of treatment, type and duration of antibiotic treatment leading up to FMT or RBT, amount of FMT/RBT material administered, FMT donor (related/unrelated), time to recurrence following clinical resolution from CDI, time to treatment failure (i.e., recurrence of CDI within the first eight weeks of treatment), and time of death. See Appendix A for definitions.

### 2.3. Outcome

The primary endpoint was clinical resolution from CDI eight weeks after FMT or RBT treatment. Clinical resolution was defined as absence of diarrhoea or a negative polymerase chain reaction test (PCR test) for *Clostridioides difficile*. Diarrhoea was defined as three or more loose or liquid defecations a day (Bristol Stool Chart 6–7). Secondary endpoints were time to recurrence after initial clinical resolution, and time to death. Recurrence was defined as diarrhoea caused by *Clostridioides difficile* (with a positive PCR test) at any timepoint after initial clinical resolution.

### 2.4. Statistical Analyses

Continuous variables are presented as mean ± standard deviation or median [interquartile range]. Baseline differences between groups were analysed with the Analysis of Variance (ANOVA) test if normally distributed or Kruskal–Wallis test if not. Categorical variables are presented as proportions and were analysed using Chi^2^ or fishers exact test as appropriate. A logistic regression analysis with covariates was conducted to adjust for predicting variables. Multiple comparisons were included in the regression analysis. Cox regression was applied to calculate hazard ratios.

Data were analysed using the statistical software R^®^ with a statistical significance level of ≤0.05 (two-sided). Patients, who died before the primary endpoint was met, were excluded in calculation of the primary endpoint and time of recurrence.

### 2.5. Treatments

The choice of treatment (FMT enema, FMT capsules or RBT) was based on a combination of the clinical decision of a physician, the patients’ preferences, and the availability of treatments. FMT enema and RBT were available in the entire five-year period 1 January 2017 to 31 December 2021, whereas FMT capsules were introduced as a treatment option from 1 September 2020. Clinical follow-up was comprised of telephone consultations by a physician from the Department of Infectious Diseases, CUHH, with expertise in treatment of CDI one week and eight weeks after treatment. Patients received either vancomycin or fidaxomicin for 4–10 days until 24–36 h prior to FMT or RBT. In 2017 to 2020, the vancomycin dose was 500 mg × 4 daily. From 1 January 2021 the vancomycin dose was reduced to 125 mg × 4 daily. The dosage of fidaxomicin remained 200 mg × 2 daily during the entire period. Treatment with proton-pump inhibitors was suspended 24 h prior to treatment with FMT capsules.

Material for FMT was either supplied by a FMT stool bank at CUHH [20,25] or Centre for Faecal Transplantation (CEFTA) at Aarhus University Hospital [26], depending on availability.

In 2017, nine patients received fresh FMT enema from close relatives. Other donors for FMT enema and capsules were unrelated and recruited from the Danish Blood Donor Corps. Both related and unrelated donors were screened and tested according to the international guidelines for organisation of FMT and donor recruitment [21,27,28]. The amount of material from related donors used in the production of one FMT enema treatment varied from 37 g to 200 g. FMT enema produced from anonymous donors and FMT capsules derived from 50 g crude non-lyophilized faecal material [26,28,29] and were subsequently stored at −80 °C.

Frozen un-lyophilized FMT capsules were ingested with apple or orange juice during a time-period of 10–15 min after 6 h of fasting. For FMT enema, both fresh FMT and FMT material from the stool bank were administered 30–40 cm rectally (or in three cases by stoma) by a white Mülly suction catheter (Ch 12). The process took 10–15 min. The patients were positioned on their left lateral side with bended hips and knees. Patients stayed in this position for 1 h. Both FMT enema and capsules were administered once as a single treatment.

Rectal bacteriotherapy (RBT) consisted of a 200 mL standardised bacterial mixture of 12 well-defined bacterial strains previously described elsewhere [17,18]. The bacterial mixture was produced at MT-MicroSearch Aps on the day of administration and transported directly from the laboratory to the Department of Infectious Diseases, CUHH. RBT was administered similarly to FMT enema but on three consecutive days.

## 3. Results

We screened 375 patients treated with FMT or RBT in the period from 1 January 2017 to 31 December 2021 for inclusion in this retrospective cohort study: 32 patients were excluded resulting in 343 included patients; 71 patients received FMT capsules (of whom 45 patients were treated with capsules produced at CUHH); 96 patients received FMT enema (of whom 71 patients were treated with FMT produced at CUHH); 176 patients received RBT. Nine patients died before meeting the primary endpoint; two in the FMT capsule group, four in the FMT enema group and three in the RBT group (Figure 1). None of these were considered related to FMT or RBT treatment.

Median follow-up time for all patients, including patients who died before the primary endpoint was met and patients with treatment failure, was 179.0 [interquartile range, IQR 173.5] days in the FMT capsule group, 64.0 [338.8] days in the FMT enema group and 123.5 [500.0] days in the RBT group. Appendix A shows an overview of the follow-up period. The mean age (64.1 ± 18.6 years) of the patients receiving FMT capsules was significantly lower than the RBT treated group (69.1 ± 17.0) (Table 1). Six patients received either FMT enema or FMT capsules for their first CDI, because of severe/fulminant or treatment refractory infection. Median duration of antibiotic treatment prior to FMT and RBT was 58.0 [54.5] days in the FMT capsule group, 46.5 [65.3] days in the FMT enema group and 81.0 [41.6] days in the RBT group. The majority of patients in the RBT group were treated with vancomycin prior to RBT. Moreover, the amount of donor material used in the production of FMT was significantly higher in the FMT enema group compared with the FMT capsule group, since nine patients in the FMT enema group received fresh FMT from related donors, deriving from 37 g to 200 g faecal material. Otherwise, there was no difference between the three groups at baseline (Table 1).

### 3.1. Clinical Resolution

The overall rate of clinical resolution was 79.7% (55/69) in the FMT capsule group, 53.3% (49/92) in the FMT enema group, and 61.8% (107/173) in the RBT group. The unadjusted odds ratio for clinical resolution was 3.45 (CI: 1.71 to 7.24) in the FMT capsule group compared with the FMT enema group, and 2.42 (CI: 1.28 to 4.84) in the FMT capsule group compared with the RBT group. There was no difference between the RBT and the FMT enema groups (odds ratio 1.09 (CI: 0.51 to 2.45)) (Appendix A). After adjusting for age, gender, CCI, number of previous CDI’s, *Clostridioides difficile* subtype, severity of current CDI episode, and the duration of antibiotic treatment leading up to FMT or RBT, the odds ratio for clinical resolution in the FMT capsule group was 3.79 (CI: 1.82 to 8.26) compared with FMT enema and 2.92 (CI: 1.49 to 6.03) compared with RBT. The adjusted model found no difference between the RBT group and the FMT enema group (Figure 2, Appendix A).

Subdividing the population into the number of CDI recurrences, we found increased treatment efficacy in the FMT capsule treated patients in the subgroups first CDI to 1st recurrence and 2nd recurrence, but no difference in treatment efficacy in patients with ≥3rd recurrences (Appendix A) compared with FMT enema and RBT. However, the confidence intervals are large, suggesting that the sample sizes are too small when dividing patients into subgroups. Moreover, because of small numbers, it was not possible to calculate a fully adjusted odds ratio.

The higher clinical resolution in the FMT capsule group was not affected when performing a sensitivity analysis including patients who died before the primary endpoint was met (data not shown), nor the year of treatment (Appendix A). Furthermore, excluding patients in the FMT enema group who received FMT from related donors, deriving from 37g to 200 g of faecal material, did not alter the results (adjusted odds ratio 4.57 (CI: 2.15 to 10.15)). Another sensitivity analysis showed no difference in efficacy between FMT produced at CEFTA or CUHH (Appendix A).

Of patients who did not achieve clinical resolution, the median time to failure was 23 [26] days in the FMT capsule group, 18 [17] days in the FMT enema group and 14 [17] days in the RBT group (no significant difference between groups).

### 3.2. Recurrence

The proportion of patients with recurrence of CDI during the first 12 months of follow-up after initial clinical resolution was 5.5% (3/55) in the FMT capsule group, 22.4% (11/49) in the FMT enema group and 20.6% (22/107) in the RBT group, corresponding to an odds ratio of 0.20 (CI: 0.04 to 0.69) in the FMT capsule group compared with the FMT enema group, and 0.22 (CI: 0.05 to 0.68) in the FMT capsule group compared with the RBT group. Figure 3 shows the cumulative incidence of recurrence within the first 12 months of follow-up in patients who initially achieved clinical resolution. Of note, all patients in the FMT capsule group who had recurrence after initial clinical resolution, had recurrence within the first 58 days of follow-up.

Adjusting for age, gender, CCI, number of previous CDI’s, *Clostridioides difficile* subtype, and the duration of antibiotic treatment leading up to FMT or RBT, we found a reduced hazard ratio for recurrence up to 12 months after initial clinical resolution from CDI in the FMT capsule group compared with the FMT enema group, as hazard ratio was 0.24 (CI: 0.06 to 0.89). Additionally, there was a reduced hazard ratio in the FMT capsule group compared with RBT, 0.26 (CI: 0.08 to 0.91). There was no difference in time to recurrence between patients treated with FMT enema or patients treated with RBT. Appendix A shows the proportion of recurrence and corresponding hazard ratios subdivided into time periods.

### 3.3. Mortality

Including patients who died before meeting the primary endpoint, 12.7% (9/71) of patients in the FMT capsule group died during the first 12 months of follow-up compared with 26.0% (25/96) in the FMT enema group and 15.9% (28/176) in the RBT group. This corresponds to an odds ratio of 0.41 (CI: 0.17 to 0.92) in the FMT capsule group compared with the FMT enema group, 0.77 (CI: 0.33 to 1.66) in the FMT capsule group compared with the RBT group, and 0.54 (CI: 0.29 to 0.99) in the RBT group compared with the FMT enema group. Figure 4 shows a Kaplan–Meier curve of survival probability, where the log-rank test was insignificant. There was no difference in unadjusted hazard ratios between the groups. After adjusting for age, gender, CCI, number of previous CDI’s, *Clostridioides difficile* subtype, severity of current CDI and the duration of antibiotic treatment leading up to FMT or RBT, there was a trend towards a reduced hazard ratio within the first 12 months after treatment in the FMT enema group compared with the RBT group (hazard ratio 0.58, CI: 0.33 to 1.01).

## 4. Discussion

In this retrospective cohort study of 343 patients with recurrent or treatment refractory CDI, we found a higher efficacy in the FMT capsules group compared with both the FMT enema and RBT groups. The adjusted odds ratio for clinical resolution from CDI was 3.79 (CI: 1.82 to 8.26) in the FMT capsule group compared with the FMT enema group, and 2.92 (CI: 1.49 to 6.03) compared with the RBT group.

Median duration of antibiotic treatment prior to FMT and RBT was different between groups. The reason for this difference might be caused by the longer waiting time for treatment with RBT, as RBT was produced outside of the treatment facility. Additionally, the majority of patients in the RBT group were treated with vancomycin prior to RBT. As vancomycin can alter the diversity of the gut microbiome, more than fidaxomicin [30], the duration of antibiotic treatment was adjusted for in the statistical models.

In a subgroup analysis dividing the population into the number CDI recurrences, the increased treatment efficacy in the FMT capsule treated patients remained in the subgroups with first CDI to 1st recurrence and 2nd recurrence, but we found no difference in efficacy in patients with ≥3rd recurrences (Appendix A) compared with FMT enema and RBT. Albeit the large confidence intervals suggest that the sample size is too small, it might indicate that patients with ≥3rd recurrences are difficult to treat, even with FMT capsules and that FMT must be given at an early stage [31].

In patients achieving clinical resolution, the risk of recurrence within the first 12 months of follow-up was significantly lower in the FMT capsule treated patient compared with both patients treated with FMT enema and RBT. We could not find any statistically significant differences in the hazard ratios for mortality within the first 12 months of follow-up.

FMT may be administered through upper or lower endoscopy, jejunal tube, rectal enema and oral capsules. While upper and lower endoscopy, especially colonoscopy, is associated with high efficacy [12,16,32], it is followed by increased healthcare cost and possible complications for patients. FMT capsules and enema are easily administered with a low rate of complications [20]. FMT enema can be performed in patients’ private homes or nursing homes and is therefore suitable in older, frail patients, who cannot ingest FMT capsules or manage treatment in a hospital setting. Previous studies have demonstrated correspondingly high efficacy of capsules compared with colonoscopy [12,33]. Our reported rate of clinical resolution following FMT capsule treatment are comparable to previously reported results of FMT capsule treatment in patients with rCDI [15].

Treatment with FMT is predominantly associated with mild, transient side effects such as abdominal discomfort, flatulence and nausea. Nevertheless, there remains a potential risk of transmitting pathogens to the recipient even with thorough donor screening [34,35]. RBT has previously been hypothesised to be a safer alternative to FMT, because it consists of a well-defined mixture of enteric, commensal bacteria, all susceptible to either ampicillin or metronidazole. However, a long-term follow-up study by Cold et al. showed no difference in hospitalisation or mortality between patients with recurrent CDI receiving FMT enema, FMT capsules and RBT [20]. Despite the suggested lower efficacy, RBT might therefore, be an alternative treatment option in patients, who cannot tolerate FMT treatment.

The optimal composition of the content in the FMT and RBT material to ensure efficacy remains to be elucidated. In relation to FMT delivered through enema, we are not aware of published papers describing certain gut microbiota taxa correlated to treatment success. Most reported studies are from treatment delivered through endoscopy. Here, engraftment of certain bacteria such as Lachnospiraceae and Ruminococcaceae has been associated with treatment success [36,37]. Furthermore, other transferred material through FMT such as bacteriophages and fungi might confer important roles in treatment success of rCDI [38,39].

In RBT, Tvede et al. has proposed that *Bacteroides* species, which account for approximately 90% of the gut microbiome, are essential in preventing recurrence of CDI [18]. Additionally, strains of *Escherichia coli*, *Paraclostridium bifermentans* and *Peptostreptococcus productus* might be of importance as they inhibit the in vitro growth of *Clostridioides difficile,* which in turn inhibits growth of the *Bacteroides* species [17].

The strengths of this study are the long follow-up period and the availability of patient information in this large cohort of patients treated for rCDI. The median follow-up was, however, shorter in the FMT enema group compared with both FMT capsule and RBT groups. This might be caused by the increased, but not statistically significant, mortality rate and rate of new CDI recurrence in the FMT enema group, as there were no patients lost to follow-up. The secondary sector in Denmark, consisting of the public hospitals, accounts for the majority of all treatments of CDI. All contacts within the secondary sector in the eastern part of Denmark, as well as time of death, are available in one single electronic health record. Moreover, microbiological test results from both the primary and secondary sectors, are recorded in a nationwide database, which was available in this study.

The limitation of the study is the lack of randomisation. The choice of treatment was based on a clinical decision by a physician, the preferences of the patients, and the availability of the three treatment options at the time of referral. Patients unable to ingest FMT capsules might have had an underlying medical condition influencing the study outcome and life expectancy. This was not adjusted for, as the reason for choice of treatment was not routinely specified in the health records. Patients in the FMT capsule group were five years younger than patients in the FMT enema and RBT group. There were, however, no statistical differences in gender, CCI, severity of current CDI, the number of previous CDI episodes, the *Clostridioides difficile* subtype, immunosuppression or treatment with proton-pump inhibitors between the three groups, and the hazards for mortality after treatment were not statistically different, suggesting that patients in the three treatment groups were alike. Nevertheless, even though it is not reflected in the CCI, there might have been a tendency to administer RBT for frail patients, because of the potential risk of transmitting pathogens via FMT. Lastly, FMT capsules were introduced as a treatment option approximately 1.5 years before end of follow-up, resulting in only 71 included patients in the FMT capsule group. A longer inclusion and follow-up period might have strengthened the study.

In conclusion, this study showed higher efficacy and a lower risk recurrence following successful treatment of rCDI in the patients treated with FMT capsules compared with patients treated with FMT enema and RBT. The hazards rate for mortality were similar in the three groups. Randomised controlled trials are needed to substantiate these results.

## Figures and Tables

**Figure 1 cells-11-03272-f001:**
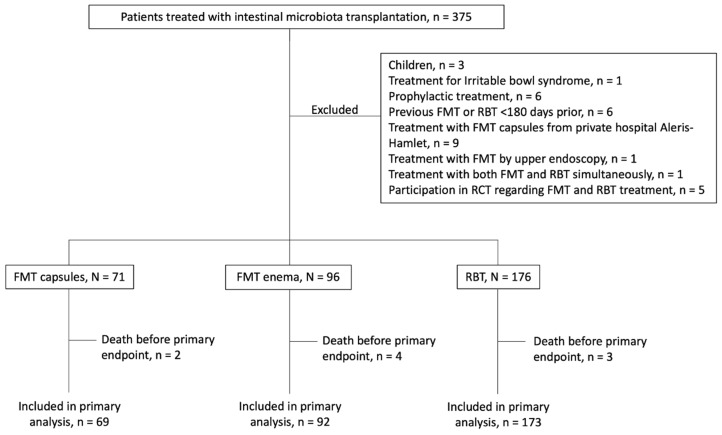
Flow chart.

**Figure 2 cells-11-03272-f002:**
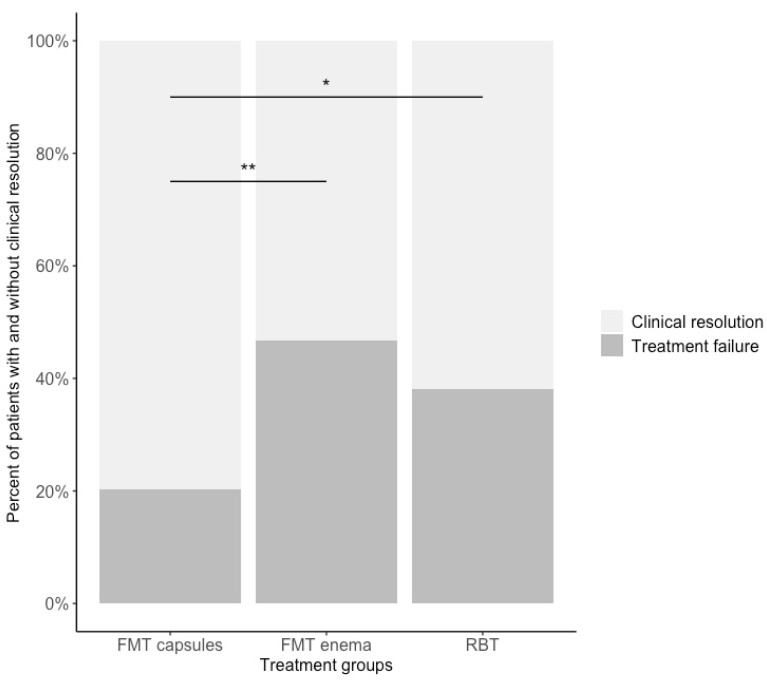
The percentage distribution of patients with and without clinical resolution in the three treatment groups. *, adjusted odds ratio *p* value < 0.01; **, adjusted odds ratio *p* value < 0.001. Odds ratios are adjusted for age, gender, CCI, number of previous CDI’s, *Clostridioides difficile* subtype, severity of current CDI episode, and the duration of antibiotic treatment leading up to FMT or RBT.

**Figure 3 cells-11-03272-f003:**
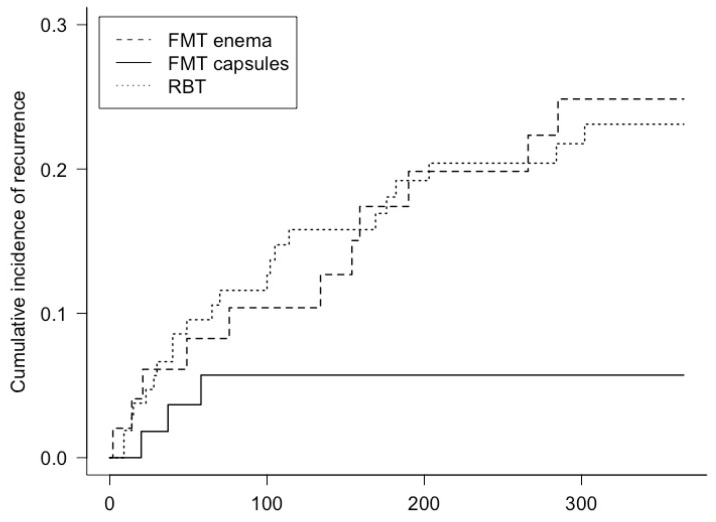
The cumulative incidence of recurrence from the time of primary endpoint (eight weeks after treatment) and during the subsequent 12 months of follow-up in patients, who initially had clinical resolution. Of note, the last event in the FMT capsule group occurred on day 58 of follow-up, causing a stagnated curve.

**Figure 4 cells-11-03272-f004:**
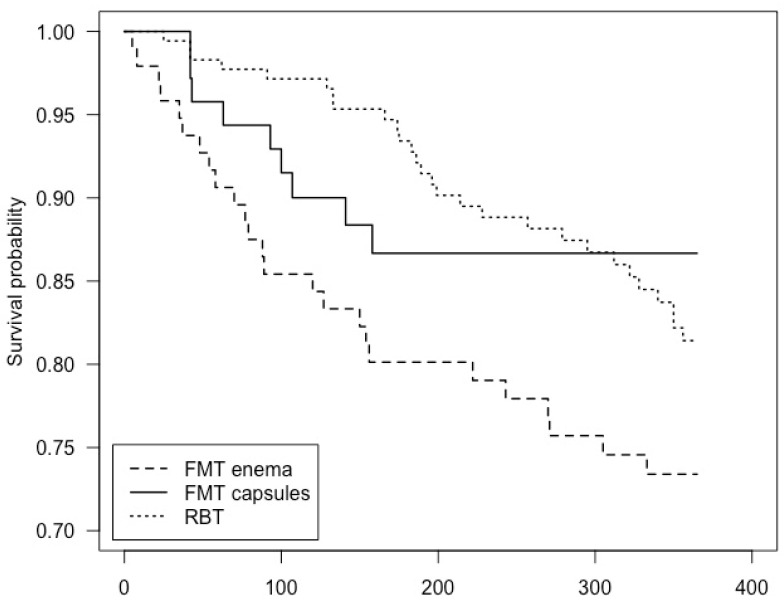
Kaplan–Meier curve of survival probability from treatment date and the subsequent 12 months of follow-up, including patients who died before meeting the primary endpoint. Of note, the last event in the FMT capsule group occurred on day 114 of follow-up, causing a stagnated curve.

**Table 1 cells-11-03272-t001:** Baseline characteristics.

	FMT Capsules N = 71	FMT Enema N = 96	RBT N = 176	*p*
Male gender, n (%)	25 (35.2)	46 (47.9)	72 (40.9)	0.14 ^a^, 0.49 ^b^, 0.32 ^c^
Age, years	64.1 ± 18.6	69.0 ± 18.1	69.1 ± 17.0	0.09 ^a^, 0.04 ^b^, 0.94 ^c^
CCI	5.0 [3.5]	5.0 [3.0]	5.0 [3.0]	0.51 ^a^, 0.47 ^b^, 0.96 ^c^
Previous CDI, n	3.0 [1.0]	2.0 [1.0]	3.0 [1.0]	0.18 ^a^, 0.06 ^b^, 0.73 ^c^
Severity				0.61 ^a^, 0.49 ^b^, 0.07 ^c^
Mild/moderate, n (%)	70 (98.6)	92 (95.8)	175 (99.4)	
Severe, n (%)	1 (1.4)	2 (2.08)	1 (0.6)	
Fulminant, n (%)	0	2 (2.08)	0	
*Clostridioides difficile* subtype				0.66 ^a^, 0.49 ^b^, 0.08 ^c^
CD027, n (%)	4 (5.6)	16 (16.7)	35 (19.9)	
Non-027, n (%)	45 (63.4)	64 (66.7)	99 (56.2)	
Not specified, n (%)	22 (31.0)	16 (16.7)	42 (23.9)	
Proton-pump inhibitor, n (%)	31 (43.7)	41 (42.7)	80 (45.5)	1.00 ^a^, 0.91 ^b^, 0.76 ^c^
Immunosuppression, n (%) *	18 (25.4)	19 (19.8)	41 (23.3)	0.50 ^a^, 0.86 ^b^, 0.61 ^c^
Previous treatment with either FMT or RBT, n (%) **	5 (7.0)	8 (8.3)	12 (6.8)	0.99 ^a^, 0.07 ^b^, 0.83 ^c^
Most recent antibiotics for CDI				1.00 ^a^, 0.054 ^b^, <0.01 ^c^
Vancomycin, n (%)	67 (95.4)	89 (92.7)	174 (98.9)	
Fidaxomicin, n (%)	3 (4.2)	5 (5.2)	2 (1.1)	
Duration of antibiotics, days	58.0 [54.5]	46.5 [65.3]	81.0 [41.6]	0.29 ^a^, <0.001 ^b^, <0.001 ^c^
Amount of faecal material, g	50.0 [0.0]	50.0 [0.0]	-	0.001 ^a^
Related donor, n (%)	0	9 (9.4)	-	0.01 ^a^
Follow-up time, days	179.0 [173.5]	64.0 [458.5]	123.5 [500.0]	0.30 ^a^, 1.00 ^b^, 0.41 ^c^

CCI, Previous CDI, duration of antibiotic treatment, amount of faecal material and follow-up time are reported as medians with [interquartile range, IQR]. Other values are reported as means ± standard deviations or number of patients (%). CCI, Charlson Comorbidity Index; CDI, Clostridioides difficile infection; FMT, Faecal microbiota transplantation; *p*, *p* value; RBT, rectal bacteriotherapy. * Two patients in the RBT group had B cell deficiency and Immunoglobulin G (IgG) deficiency, respectively. One patient in the FMT enema group had Mannose-binding Lectin (MBL) deficiency. The remaining patients were treated with immunosuppressive medication. ** Previous treatment six months or more, prior to current treatment. ^a^ FMT capsules versus FMT enema, ^b^ FMT capsules versus RBT, ^c^ RBT versus FMT enema.

## Data Availability

The data supporting all the figures and tables in the published article are not publicly available due to restrictions. The authors do not have permission to share the data.

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
