# Peer review of "The Efficacy of Faecal Microbiota Transplant and Rectal Bacteriotherapy in Patients with Recurrent Clostridioides difficile Infection: A Retrospective Cohort Study"

_cells, 2022, doi:10.3390/cells11203272_

Round 1
Reviewer 1 Report
Svensson et. al should be commended for their extensive analyses of the studied cohort. My comments on their manuscript are below.
Whilst useful to include non-FMT 'FMT-like' preparations as treatment options for CDI in this paper (as it is likely to be where the field goes eventually), I do not feel that the choice of 'intestinal microbiota transplantation' (IMT) is an appropriate 'catch-all' term for FMT and RBT, as some individuals use IMT as a substitute for FMT in the literature. Therefore, I would suggest they refrain from using this term throughout the manuscript, and come up with another descriptor to avoid ambiguity.
I would also request that data including individuals treated with FMT via the nasogastric/jejunal route or colonoscopy is included, if available. It is not clear why this has been excluded. From a global perspective, FMT is rarely administered via enema (for CDI), and I don't believe RBT is given outside of Denmark, so in its current form the manuscript has limited broader applicability.
Supplementary material - I think it would be good to clarify what is meant by 'previous CDI episodes' - if this is indeed previous episodes, then only 3+ would be in line with most current guidelines for rCDI (i.e. 2+ recurrences). The lower success rate with capsules if treating as per this guidance, compared to earlier in illness journey, is a really important point to highlight and of marked clinical relevance. If indeed capsules only have a success rate of 67%, this is below what is quoted for endoscopic methods, and an important finding of the study.
There is a large, albeit not statistically significant, difference in the median follow-up of patients across the treatment arms - could the authors offer an explanation for this? This is important as long duration of follow-up is described as one of the strengths of the study.
I believe that the referenced trial [18] results infer that RBT is inferior to FMT delivered via enema.
Minor
Line 42 (and elsewhere) - I believe it would be helpful to clarify if the capsules are lyophilised or not (I presume not).
Line 59 (and elsewhere) - million should be shortened to 'mill' or 'M'
Line 72 - it would be good to qualify why the FMT capsules produced at the private hospital were excluded
Line 156 - grammatical error - 'were' should be 'was'
Line 180 - spelling error - 'tree' should be 'three'
Line 187 - spelling error - 'Materiel' should be 'Material'
Supplementary - Cell counts should be written as 2x10^6/L etc.
References - there appear to be some formatting errors e.g. ref [21]
Reviewer 2 Report
In this retrospective cohort study, the authors aimed to evaluate the efficacy of three treatment modalities for recurrent Clostridioides difficile (rCDI) infection including capsules, enema, and rectal bacteriotherapy (RBT). The topic is of interest to the readers of Cells and requires continued investigation in order to optimize fecal microbiota transplantation (FMT) treatment. Below are the reviewers’ comments that authors should address to improve the quality of the publication.
-Although the outcomes of the study are discussed, the treatments themselves should be discussed further. A more in-depth discussion is required to compare RBT vs. capsule vs. enema as a treatment. The advantages/disadvantages should be compared and discussed further. What are the most common gut microbiota that can lead to a successful enema vs. a successful RBT, etc.?
-Today it’s more critical than ever to discuss the importance of all organisms involved (i.e., viruses, fungi, and protozoa) and the multifaceted system which can affect gut microbiota and treatment efficacy. This should also be discussed in the introduction or discussion section.
-Overall, table 1 is poorly discussed within the text of the results section: In line 156 of the results section, how were the means compared? What statistical test was used here and what are the p-values? In line 157, the same could be said. How was the duration of the antibiotic treatment prior to FMT and RBT compared? What statistical test was used here and what are the p-values? In line 168, stating that there was no significant difference between RBT versus enema-what’s the p-value? What about the other statistically significant findings in table 1 that weren’t mentioned in the results section such as recent antibiotics for CDI, duration, amount of faecal matter, etc. These should be mentioned in the results section and discussed further in the discussion as well.
Round 2
Reviewer 1 Report
No further comments.